# Prevalence and Genetic Analysis of Resistance Mechanisms of Linezolid-Nonsusceptible Enterococci in a Tertiary Care Hospital Examined via Whole-Genome Sequencing

**DOI:** 10.3390/antibiotics11111624

**Published:** 2022-11-14

**Authors:** Yuxin Hu, Dongju Won, Le Phuong Nguyen, Kennedy Mensah Osei, Younghee Seo, Junglim Kim, Yoonhee Lee, Hyukmin Lee, Dongeun Yong, Jong Rak Choi, Kyungwon Lee

**Affiliations:** 1Department of Laboratory Medicine and Research Institute of Bacterial Resistance, Yonsei University College of Medicine, Seoul 03722, Republic of Korea; 2Brain Korea 21 PLUS Project for Medical Science, Yonsei University, Seoul 03722, Republic of Korea; 3Cedars-Sinai Medical Center, Los Angeles, CA 90048, USA; 4Department of Laboratory Services, Tamale Teaching Hospital, Tamale 41965, Ghana; 5Seoul Clinical Laboratories Academy, Yongin 16954, Republic of Korea

**Keywords:** linezolid resistance, whole-genome sequencing, optrA, poxtA2–cfr(D) co-harboring, *Enterococcus* species

## Abstract

(1) Background: Linezolid plays an important role in the treatment of invasive infections caused by vancomycin-resistant enterococci after its introduction to clinical practice. However, a detailed examination of linezolid-nonsusceptible enterococci (LNSE) is required. In this study, we attempted to analyze the mechanisms of LNSE strains isolated from a tertiary care hospital. (2) Methods: From 2019 to 2020, 18 *Enterococcus faecalis*, 14 *E. faecium,* and 2 *E. gallinarum* clinical isolates were collected at Severance Hospital. Agar dilution was performed to evaluate precise linezolid minimum inhibitory concentrations (MICs). Short-read whole-genome sequencing (WGS) was used to analyze resistance determinants. (3) Results: The presence of the *optrA* gene was likely the primary resistance mechanism in these three species, typically demonstrating a MIC value of 8 μg/mL. The co-existence of the *cfr(D)* and *poxtA2* gene was the second major mechanism, primarily predicting a phenotype showing intermediate susceptibility to linezolid. G2576U mutation on 23S rRNA was only found in *E. faecium*; it mediated the most significant increase in linezolid MIC. (4) Conclusion: This is the first report examining *poxtA2–cfr(D)* co-harboring clinical enterococcal isolates in Korea and demonstrating the *poxtA EF9F6*-harboring clinical *E. gallinarum* strain worldwide. The comparison with resistance-gene-containing fragments in the isolates obtained from different countries and different sources demonstrated the spread of linezolid-resistance genes and suggested the possibility of foodborne transmission.

## 1. Introduction

*Enterococcus* species have been recognized as one of the major pathogens responsible for many infections including healthcare-associated infections in humans for decades. In addition to their intrinsic resistance to a broad spectrum of antibiotics, they can also acquire plasmids and other genetic elements that harbor antimicrobial-resistance genes [1], rendering them multi-drug-resistant pathogens, which might be difficult to treat even with vancomycin. Linezolid, the first oxazolidinone antibacterial agent approved for commercial use by the US Food and Drug Administration (FDA) in 2000, has been considered a useful therapeutic option for various infections caused by Gram-positive bacteria. However, since linezolid was introduced for clinical use, the presence of linezolid-nonsusceptible enterococci (LNSE) has continuously been reported over the years [2].

Several mechanisms responsible for decreased linezolid susceptibility have been reported. Nontransferable mechanisms include mutations on the 23S rRNA-binding site (e.g., G2576U, G2447U, and G2504A) and alterations in ribosomal proteins, such as L3, L4, and L22 [3]. Moreover, the acquisition and horizontal transfer of the genes responsible for plasmid-mediated resistance (e.g., the *cfr* gene family, *optrA*, and *poxtA*) has also been increasingly reported in linezolid-resistant enterococci from different sources and countries.

The *cfr* gene was initially discovered in *Staphylococcus sciuri* from bovine [4]. It encodes a 23S rRNA methyltransferase and confers cross-resistance against oxazolidinones, phenicols, lincosamides, pleuromutilins, and streptogramin A antibiotics (PhLOPS_A_ phenotype) [5]. To date, four variants of the *cfr* gene have been described, namely *cfr(B)*, *cfr(C)*, *cfr(D)*, and *cfr(E)*. However, *cfr(C)* and *cfr(E)* have not been reported in *Enterococcus* species yet, and *cfr(B)* and *cfr(D)* do not confer linezolid resistance when they are the only resistance mechanism expressed in an enterococcal background [6,7,8]. 

The *optrA* gene, encoding an F lineage protein of the ATP-binding cassette (ABC) protein superfamily, confers transferable resistance to oxazolidinones and phenicols via ribosomal protection. The gene was initially discovered in China in animal and human isolates [9] and was then widely reported in the American continent [10] and Europe [11,12]. 

In 2018, a novel transferrable oxazolidinone resistance gene *poxtA* was identified in a clinical methicillin-resistant *Staphylococcus aureus* strain isolated in Italy [13]. It shares a 32% sequence identity with the *optrA* gene and has a similar working mechanism. The presence of the *poxtA* gene has been reported in *Enterococcus* species isolated globally [14,15,16] and was first reported in Korea in 2020 [17]. A new *poxtA2* variant resulting in resistance to linezolid was identified in a human strain of *E. gallinarum* [18] and the occurrence of the conjugative plasmid pV386 co-carrying *cfr(D)* and *poxtA2* genes was subsequently found in *E. faecalis* and *E. casseliflavus* from swine manure [19]. Very recently, the *poxtA EF9F6* gene has been newly reported to produce decreased susceptibility to linezolid [20].

Although these linezolid resistance mechanisms have been commonly reported worldwide, relatively few reports have been published in Korea. The carriage rate has been relatively low. In a three-year survey, 0.23% of clinical *E. faecalis* isolates harbored the *optrA* gene, and no *cfr*-like genes were detected [21]. In a collection of 135 strains of *E. faecalis* and *E. faecium*, the overall frequency of G2576U mutation on 23S rRNA was 10.4% [22]. Among 389 *E. faecium* isolates, only 2 isolates had G2576U mutation, and no other resistance determinants were found [23]. Therefore, to supplement the evidence of the clinical prevalence of linezolid resistance in Korea, we used enterococcal clinical isolates from a tertiary care hospital. In the present study, the prevalence of linezolid-resistance mechanisms was determined after whole-genome sequencing (WGS), and the genetic environment of each determinant was investigated.

## 2. Results

### 2.1. Evolution of Linezolid Nonsusceptibility in Severance Hospital over the Past Five Years

The five-year linezolid-susceptibility surveillance data of Severance Hospital are presented in Table 1. Linezolid resistance increased by 3.3% (from 3.4% to 6.7%) in *E. faecalis* and 1.5% (from 3.5% to 5.0%) in *E. faecium*. The increase in the proportion of *E. faecalis* strains demonstrating intermediate resistance to linezolid (from 0.5% to 2.6%) was greater than that of *E. faecium* (from 0.1% to 1.4%). Overall, a higher proportion of linezolid-nonsusceptible *E. faecalis* (LNSEFA) strains than that of linezolid-nonsusceptible *E. faecium* (LNSEFM) strains was observed.

### 2.2. Linezolid Minimum Inhibitory Concentration (MIC) Distribution and Resistance Mechanisms

While the proportion of LNSEFAs was higher than that of LNSEFMs, LNSEFMs demonstrated overall higher MICs (Table 2). A MIC of 8 μg/mL was observed in 77.8% of LNSEFA, and all the strains carried the *optrA* gene. All the *E. faecalis* strains demonstrating intermediate resistance (MIC = 4 μg/mL) carried both *poxtA2* and *cfr(D)* genes. The resistance mechanisms of LNSEFM were more diverse, and the MIC values were more scattered. The two LNSEFM strains with a MIC of 32 μg/mL and one with a MIC of 16 μg/mL were attributed to the G2576U mutation on 23S rRNA. Similar to LNSEFA, the *optrA* genes in LNSEFMs were associated with a MIC of 8 μg/mL. An LNSEFM strain carrying the *cfr(D)* gene alone showed a MIC of 8 μg/mL; however, when the strain co-harbored the *poxtA2* gene, a tendency of intermediate susceptibility was observed. In addition, four LNSEFM strains did not have mutations on the linezolid ribosomal binding site and did not carry any known linezolid-resistant genes. As for the *E. gallinarum* isolates carrying *poxtA EF9F6* genes, the MICs of both isolates were 8 μg/mL. Overall, the presence of *optrA* was the most common mechanism (50.0%), and the associated MIC was typically 8 μg/mL. The co-existence of *poxtA2* and *cfr(D)* was the second common mechanism (14.7%), and the associated MIC was 4 μg/mL, except for one strain with a MIC of 8 μg/mL. G2576U mutation was not found in LNSEFA. However, it was the primary reason for the significantly higher MIC values of LNSEFM; 75% of LNSEFMs with G2576U mutation showed MICs from 8 μg/mL to 32 μg/mL.

### 2.3. Genetic Environment of Different Resistance Determinants

Resistance gene-harboring regions were divided into four groups according to their characteristics, such as the *cfr(D)* gene, RNA-polymerase-encoding gene, and ferredoxin-coding gene. The resistance genes of strains EFM 30, EFM 31, EFA 39, EFA 62, and EFA 64 were located on the plasmids.

Group I included all the *cfr(D)*-positive isolates (four *E. faecium* and three *E. faecalis* strains). The resistance gene-involving contigs of strains EFM 30, EFM 31, EFA 39, EFA 62, and EFA 64 were highly similar, co-harboring a *poxtA2* gene upstream. It was confirmed that both the *cfr(D)* and *poxtA2* genes were located on the plasmid. This was also evidenced by the presence of the *repR* gene 1495 bp downstream of the *cfr(D)* gene. Two contigs from *E. faecium* chromosome carrying the *cfr(D)* gene alone were also observed, which showed a GMP synthase gene downstream of the *cfr(D)* gene in the opposite direction (Figure 1). 

Group II included five *E. faecalis* strains (EFA 35, EFA 3, EFA 63, EFA 20, and EFA 26) and two *E. faecium* strains (EFM 6 and EFM 2) (Figure 2a). These strains showed a ferredoxin-coding gene downstream of the *optrA* gene in the same direction. Except for EFM 2, a *fexA* gene that mediates resistance to chloramphenicol and florfenicol was located 742 bp upstream of the *optrA* gene, and a 23S rRNA (adenine (2058)-N (6))-dimethyltransferase-encoding gene *ermA* was located 1417 bp downstream of the *optrA* gene in the opposite direction. Due to the contig length limitation, more information about the environment of the *optrA* gene of EFM 2 could not be obtained. However, it was found that EFA 599799 shared high similarity to the EFM2 *optrA*-containing contig and additionally carried a gene encoding a NAD(P)H oxidoreductase.

Group III included EFA 25 alone (Figure 2b). These strains demonstrated a unique genetic environment in this study, carrying a gene encoding RNA polymerase upstream of the *optrA* gene. 

Group IV included ten *optrA*-harboring isolates, one of which was *E. faecium* and the remaining nine were *E. faecalis*. They were associated with Tn554 components, followed by *fexA*, *optrA*, ribonuclease J1-encoding gene, and two heterodimeric efflux ABC transporter genes *efrA* and *efrB*. EFA 10 and EFA 28 both showed an insertion of approximately 900 bp between the Tn554 and the *fexA* gene (Figure 2c). For a detailed examination of Tn554 and Tn554-containing gene fragments, EFM 5 and EFA 4 were selected as representative isolates for comparison with the Tn554 reference sequence (GenBank accession no. X03216.1) (Figure 3). The original Tn554 comprises three transposase genes *tnpA*, *tnpB*, and *tnpC*, *ant(9)-Ia* (also known as the *spc* gene), and the erythromycin resistance methylase-encoding gene *ermA*. In EFM 5, an *optrA*-containing fragment of approximately 7800 bp replaced the *ant(9)-Ia* gene. Although EFM 5 harbored the Tn554 transposase-encoding genes *tnpA*, *tnpB*, and *tnpC*, when compared to the Tn554 reference sequence, the sequence similarity only reached 70%.

The *poxtA EF9F6* gene, which was first reported very recently, was found in four strains in our study: EGA 11, EGA 12, EFM 2, and EFM17. Due to the limited contig length, the genetic environment of the *poxtA EF9F6* gene in EFM 2 and EFM17 did not yield sufficient information. The *poxtA EF9F6* gene of EGA 11 and EGA 12 shared the same genetic environment. It was bracketed by IS1380-like transposase-coding gene upstream and downstream.

### 2.4. Clonal Relatedness and Other Molecular Features

Two novel *E. faecalis* sequence types (ST) (ST1166 and ST1167) and four alleles (aroE113, gdh107, xpt102, and yqiL105) were identified for the first time. Closely related isolates were divided into six groups, as shown in Figure 4. In any of Group 1, Group 2, Group 3, and Group 4, despite the fact that they were isolated from different patients in different departments, they had close clonal relatedness and a very high degree of genome similarity, suggesting nosocomial infections caused by LNSEFA. All the strains in Group 5 and Group 6 were also clonal. They were all isolated from inpatients in the Hematology Department, revealing the existence of LNSEFM in the ward at least from June 2019 to November 2019. 

A consistency in the resistance mechanisms and MIC pattern in *E. faecalis* was found: 93.3% (14/15) of the isolates carrying the *optrA* gene showed MICs of 8 μg/mL and the isolates co-harboring *poxtA2* and *cfr(D)* showed MICs of 4 μg/mL (Figure 4a). Compared with *E. faecalis*, the clonal relatedness between *E. faecium* isolates was high (all strains belonged to CC17); however, the resistance mechanisms were complex, and the MIC values were scattered (Figure 4b). For both *E. faecalis* and *E. faecium*, it can be inferred that linezolid susceptibility was not remarkably related to ST and was only associated with the resistance mechanism. 

## 3. Discussion

In Korea, relatively few reports have been published examining linezolid resistance in clinical *E. faecalis* isolates [21]. However, some trends have been observed from other surveillance data examining *E. faecium*. According to a Kor-GLASS report published in 2017, 0% of *E. faecium* strains were linezolid-resistant, while approximately 3% of the strains showed intermediate susceptibility to linezolid [24]. In our study, the proportion of linezolid-resistant *E. faecium* was higher, and the proportion of *E. faecium* strains showing intermediate susceptibility to linezolid was reduced. Even though the difference in linezolid susceptibility can be influenced by many factors, this shift from intermediate susceptibility to resistance may be attributed to the continued widespread use of linezolid [25] and the high adaptability of enterococci to antibiotic pressure. Hence, there is an urgent need for constant LNSE monitoring and regulation for each hospital through wide surveillance networks.

Before 2012, the primary oxazolidinone resistance mechanism known in both *E. faecalis* and *E. faecium* involved mutations on 23S rRNA and ribosomal proteins, which could be spread by clonal dissemination [3]. However, since the discovery of plasmid-mediated linezolid resistance, the influence on the resistance mechanism has demonstrated a gradual shift in the presence of *optrA, poxtA*, and *cfr* genes. The presence of *optrA* has now become the dominant mechanism of resistance in the strains isolated from 2014 to 2016 [26]. This trend has also been demonstrated in a study on the clinical *E. faecalis* isolates obtained in Korea; all linezolid-nonsusceptible isolates were *optrA*-positive [21]. In this study, the presence of G2576U mutation on 23S rRNA led to a more significant increase in linezolid MIC values than other mechanisms. The strong influence of G2576U mutation on the increase in the MIC value has also been reported in Korea, where the MIC values of both linezolid-resistant *E. faecium* isolates were higher than 64 μg/mL [23]. The four linezolid-resistant strains EFM 13, EFM 22, EFM 29, and EFM 46 did not demonstrate any existing resistance mechanism. A similar conclusion was drawn in a Korean study conducted in 2017 that no corresponding mechanism was found in two-thirds of linezolid-resistant *E. faecium* strains [22].

When comparing the genetic environment of resistance genes, not only the strains isolated in this research but also published sequences were included if a high similarity was found. A good consistency was observed in the sequence of the *optrA* gene between previous studies and this study, but the genetic environment of *optrA*-flanking regions displayed relatively low homology. Two *E. faecalis* strains examined in an international surveillance study (EFA 838523 and EFA 599799) [26] carrying a ferredoxin-coding gene downstream of the *optrA* gene belonged to Group II. A Korean foodborne *E. faecalis* isolate [27] and a Chinese *E. faecalis* isolate of human origin [28] harboring the *optrA* gene were categorized as Group III. As for EFA4 and other strains in Group IV, the *fexA*–*optrA* fragment was inserted at the end of Tn554. One possible explanation is that the resistance gene-containing fragment was acquired in the past few generations, and such a result was obtained after continuous gene rearrangement.

Thus far, no *cfr*-carrying clinical enterococcal isolates have been reported in Korea. However, several *cfr*-mediated linezolid resistance mechanisms have been identified in the *S. aureus* strains isolated from pig carcasses [29] and the *E. faecalis* strains isolated from food-producing animals [30], indicating the possibility of foodborne transmission. The co-location of *optrA* and *cfr* has been reported in human *E. faecium* isolates observed in Ireland [31] and Italy [32], providing evidence of their prevalence worldwide. The gene fragments in Group I carrying both *poxtA2* and *cfr(D)* are highly similar to those of *E. faecalis* V386 isolated from Italy (NCBI Reference Sequence: NZ_MZ603802.1), with a *repR* gene located downstream. Compared with the 37 coding DNA sequences (CDSs) of v386, 34 CDSs of the two *E. faecium* strains were the same, and 13 CDSs of the three *E. faecalis* strains were the same. Hence, it can be inferred that the resistance was transmitted by the plasmid, even across species. A study has revealed the co-existence of *poxtA2* and *cfr(D)* gene in two *E. faecalis* strains with different genetic environments. Both strains were isolated in Korea in 2018 [30]. 

As confirmed by MobileElementFinder, strains EFM 30, EFM 31, EFA 39, EFA 62, and EFA 64 had genes located on plasmids and were at risk of widespread horizontal transmission. However, one of the limitations of this study was that we only used the second-generation sequencing method. If MobileElementFinder did not show that the resistance gene was located on the plasmid, it could not be concluded that the resistance gene was located on the chromosome. Long-read sequencing methods such as Oxford Nanopore MinION could be applied to help with defining gene location and plasmid reconstruction.

In this study, only one of the strains carrying both *poxtA2* and *cfr(D)* genes (EFM 30) was resistant to linezolid, while the other four strains only had intermediate resistance. According to previous studies, a single *poxtA2* gene was responsible for the resistance to linezolid in enterococci with a MIC of 8 μg/mL [18]. That is to say, lower linezolid MICs could be observed in the strains co-harboring *poxtA2* and *cfr(D)* genes in comparison to those strains carrying only the *poxtA2* gene. On the one hand, there is no one-to-one correspondence between bacterial antibiotic-resistant phenotypes and genotypes. A similar phenomenon has been reported in other studies. A pig-origin *E. casseliflavus* strain co-harboring *poxtA2* and *cfr(D)* gene has been reported to present a MIC of 8 μg/mL [33]. The same research group later reported a *poxtA2*–*cfr(D)* co-harboring plasmid pV386 in *Enterococcus* species. Of the three strains carrying pV386, two had MICs of 4 μg/mL and the other had a MIC of 2 μg/mL [19], which means that none of the three strains was resistant to linezolid. In another study of enterococci isolated from pigs, one *E. faecalis* strain was reported that carried both *poxtA2* and *cfr(D)* genes. It was susceptible to linezolid with a MIC of 2 μg/mL [34]. On the other hand, the phenotype of resistance genes varies when expressed in different species. For example, the *cfr(D)* gene alone generally does not confer linezolid resistance when expressed in an enterococcal background but ﻿confers linezolid resistance when expressed in *Escherichia coli* [8]. To date, not much research has been conducted on *poxtA2*–*cfr(D)* co-harboring enterococci, and follow-up studies are needed to specifically explain the reason why low linezolid MICs (≤4 μg/mL) were still observed in a strain co-harboring two linezolid-resistant genes.

## 4. Materials and Methods

### 4.1. Bacterial Strains 

All the enterococcal isolates were recovered from 2016 to 2020 from clinical specimens collected from patients visiting or admitted at Severance Hospital, a tertiary-care hospital with 2437 beds in Seoul, South Korea. The clinical samples were very diverse, including urine, wound secretions, blood, catheter tips, tissue, bile, bronchoalveolar lavage, and peritoneal fluid. Species identification was performed using a MALDI-TOF mass spectrometry instrument (Microflex^®^ LRF, Bruker Daltonik, Bremen, Germany). Antimicrobial susceptibility was determined by Vitek 2 automated system (bioMériux, Marcy l’Étoile, France) with AST-P600 cards (bioMériux). All linezolid-nonsusceptible enterococcal isolates were confirmed through disk diffusion (Thermo-Fischer Scientific, Waltham, PA, USA) or antibiotic gradient diffusion methods (bioMériux). The results of the antimicrobial susceptibility testing were interpreted according to the Clinical and Laboratory Standards Institute (CLSI) Guidelines, the 32nd edition [35]. Linezolid susceptibility data were retrieved from the laboratory information system for analysis. 

To investigate linezolid resistance mechanisms in the clinical enterococcal isolates, from June 2019 to June 2020, one or two *E. faecalis* and *E. faecium* were randomly selected from the linezolid-nonsusceptible isolates collected each month. Two rarely recovered *E. gallinarum* were also analyzed for resistance mechanisms for linezolid nonsusceptibility. In total, 18 *E. faecalis*, 14 *E. faecium*, and 2 *E. gallinarum* were tested for further investigation, including MIC determination and whole-genome sequencing (Appendix A).

### 4.2. Linezolid Susceptibility Testing

In order to determine the minimum inhibitory concentrations (MICs) against linezolid among the 34 isolates selected for further investigation, Agar dilution was used to reconfirm the MIC of the selected strains following CLSI Guideline M7, 11th edition [36]. The linezolid powder used to prepare agar plates was from Dong-A Pharmaceutical (Seoul, South Korea). The MIC range tested was from 0.25 μg/mL to 64 μg/mL. All MICs were interpreted according to CLSI M100-ED32. The isolates were determined as susceptible to linezolid when MIC ≤ 2 μg/mL, intermediate resistant to linezolid when MIC = 4 μg/mL, and resistant to linezolid when MIC ≥ 8 μg/mL. The isolates for which the linezolid MICs were greater than or equal to 4 μg/mL were defined as linezolid-nonsusceptible. *E. faecalis* ATCC 29212 and *S. aureus* ATCC 29213 served as quality control (QC) strains.

### 4.3. Whole-Genome Sequencing and Bioinformatic Analysis

Bacterial DNA was extracted using GenElute™ Bacterial Genomic DNA Kits (Sigma-Aldrich, St. Louis, MO, USA). The sequencing library was prepared using a Twist Library Preparation EF Kit (Twist Bioscience, South San Francisco, CA, USA). WGS was performed on an Illumina NovaSeq 6000 sequencing platform (Illumina, San Diego, CA, USA) to yield a de novo assembly generated by Unicycler v0.4.6 [37]. The average sequencing coverages were ≥110× for all genomes. The assembled sequences have been deposited on NCBI under the BioProject ID PRJNA892662 (Appendix A). The ST was identified by using pubMLST (https://pubmlst.org/, accessed on 1 February 2021). Resistance gene analysis was performed using ResFinder 4.1 (http://cge.cbs.dtu.dk/services/ResFinder, accessed on 1 February 2021) [38] and BLAST (https://blast.ncbi.nlm.nih.gov/Blast.cgi, accessed on 1 February 2021). The location of resistance genes was investigated via MobileElementFinder v1.0.3 (https://cge.food.dtu.dk/services/MobileElementFinder/, accessed on 1 February 2021). Two common mutations on 23S rRNA, G2576U and G2505A, were screened by LRE-finder 1.0 (https://cge.cbs.dtu.dk/services/LRE-finder, accessed on 1 February 2021) [39]. Other mutations on the linezolid ribosomal binding site were confirmed through ribosome sequence alignment using MEGAX [40]. One *E. faecalis* strain and one *E. faecium* strain identified as linezolid-susceptible in this study were sent for WGS and used as reference sequences. The fragments containing the resistance genes were selected for annotation. Genome annotation was performed by RAST 2.0 (https://rast.nmpdr.org/rast.cgi, accessed on 1 February 2021) [41]. The annotated genome was then visualized using SEED Viewer version 2.0 (http://pseed.theseed.org/, accessed on 1 February 2021). Phylogenetic trees were created by Type Strain Genome Server (https://tygs.dsmz.de/, accessed on 1 October 2022) with default parameters.

## 5. Conclusions

In conclusion, this study is the first to report the co-existence of *poxtA2* and *cfr(D)* genes in clinical *E. faecalis* and *E. faecium* isolates in Korea and *poxtA EF9F6*-harboring clinical *E. gallinarum* globally. The similarities in the genetic context between Korean clinical isolates and the isolates from other countries and resources strongly indicate the spread of linezolid-resistant determinants between countries and the possibility of foodborne transmission. A reasonable explanation for four linezolid-resistant *E. faecium* strains with unknown mechanisms remains a problem that should be examined in future studies.

## Figures and Tables

**Figure 1 antibiotics-11-01624-f001:**
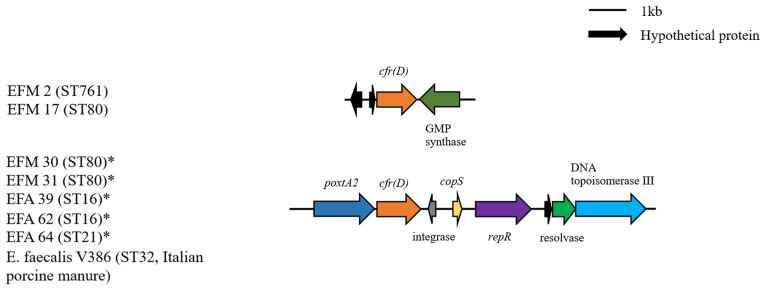
Genetic context of *poxtA2* and *cfr(D)* genes. The genes and orientation are indicated by labeled arrows, and the positions are aligned with the *cfr(D)* gene used as the benchmark. Isolates termed EFM belong to *E. faecium* and isolates termed EFA belong to *E. faecalis*. The asterisk * indicates that the contig was located on the plasmid. The nucleotide sequence of the *cfr(D)* gene was 99.9% identical to NG067192. The nucleotide sequence of the *poxtA2* gene was 100% identical to NZ_MZ603802.

**Figure 2 antibiotics-11-01624-f002:**
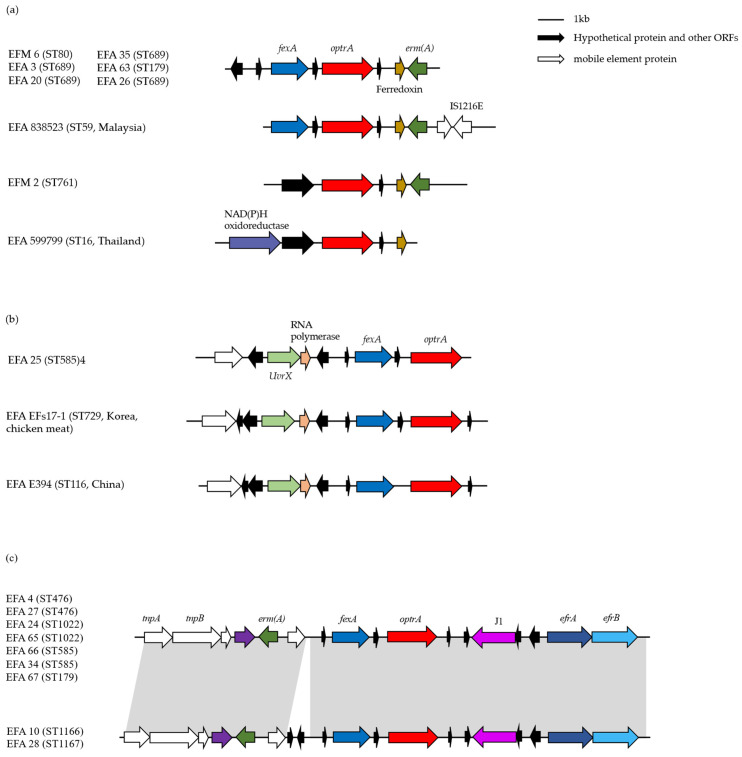
Genetic context of the *optrA* gene. The genes and orientation are indicated by labeled arrows, and the positions are aligned with the *optrA* gene used as the benchmark. Isolates termed EFM belong to *E. faecium*; isolates termed EFA belong to *E. faecalis*; isolates termed EGA belong to *E. gallinarum*: (**a**) contigs with a ferredoxin-coding gene located downstream of the *optrA* gene. The *optrA* gene of EFM 6, EFA 3, EFA 20, EFA 26, EFA 35, EFA 63, EGA 11, and EGA 12 were 100% identical to KX620942. The *optrA* gene of EFM 2 was 100% identical to ALMZ01000078; (**b**) contigs with an RNA-polymerase-encoding gene located upstream of the *optrA* gene; (**c**) contigs with Tn554 components. The gray area between the first and the second map denotes >69% sequence identity (calculated by Easyfig https://mjsull.github.io/Easyfig/, accessed on 1 November 2020). The *optrA* gene of EFA 4, EFA 10, EFA 24, EFA 27, EFA 28, EFA 34, EFA 65, EFA 66, and EFA 67 were 100% identical to KT862783. The *optrA* gene of EFM 5 was 100% identical to KX620936.

**Figure 3 antibiotics-11-01624-f003:**
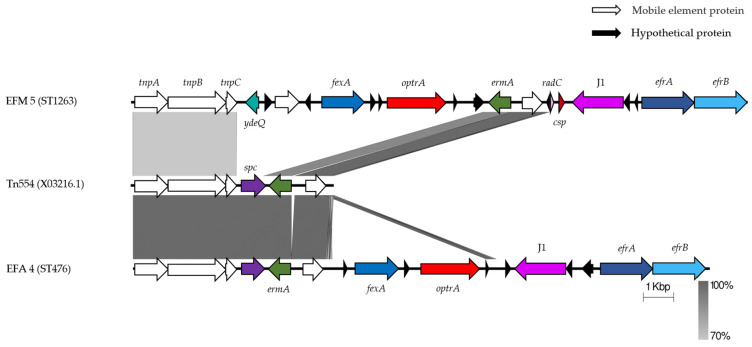
Comparison of Tn554 reference sequence and representative Tn554-related isolates, visualized by Easyfig. The genetic environment of the *optrA* gene. The genes and orientation are indicated by labeled arrows, and the positions are aligned with the *tnpA* gene used as the benchmark. Isolates termed EFM belong to *E. faecium* and isolates termed EFA belong to *E. faecalis*.

**Figure 4 antibiotics-11-01624-f004:**
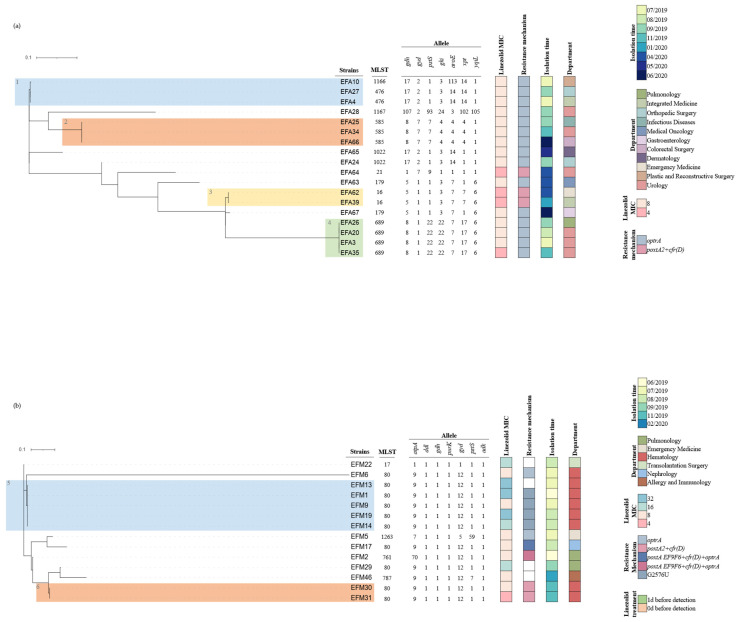
Phylogenetic tree and heatmap of (**a**) *E. faecalis* and (**b**) *E. faecium*. Colored cells represent the presence of a characteristic, and blank cells represent the absence of a characteristic.

**Table 1 antibiotics-11-01624-t001:** Linezolid susceptibility of E. faecium and E. faecalis isolated at Severance hospital from 2016 to 2020.

Year	*E. faecalis*				*E. faecium*			
	No. Isoltaes	S ^a^	I ^b^	R ^c^	No. Isoltaes	S	I	R
2016	1536	96.1%	0.5%	3.4%	2121	96.4%	0.1%	3.5%
2017	1505	95.3%	1.1%	3.7%	2407	97.6%	0.2%	2.2%
2018	1716	95.4%	1.4%	3.2%	2558	96.9%	0.4%	2.7%
2019	1544	94.2%	1.2%	4.7%	2606	95.6%	0.7%	3.8%
2020	1366	90.7%	2.6%	6.7%	2559	94.6%	0.4%	5.0%

MICs were interprettered according to the 31st edition of the Clinical and Laboratory Standards Institute (CLSI) criteria. ^a^ S refers to susceptible to linezolid. Strains are determined as S when MIC ≤ 2 μg/mL. ^b^ I refers to intermediate resistance to linezolid. Strains are determined as I when MIC = 4 μg/mL. ^c^ R refers to resistant to linezolid. Strains are determined as R when MIC ≥ 8 μg/mL.

**Table 2 antibiotics-11-01624-t002:** Linezolid MIC and resistance mechanisms of linezolid-nonsusceptible enterococci.

MIC (μg/mL)	4	8	16	32
***E. faecalis* (n = 18)**	**4 (22.2)**	**14 (77.8)**	**0**	**0**
*optrA* (n = 15, 83.3%)	1 (6.7)	14 (93.3)		
*cfr(D) + poxtA2* (n = 3, 16.7%)	3 (100)			
***E. faecium* (n = 14)**	**1 (7.1)**	**7 (50.0)**	**3 (21.4)**	**3 (21.4)**
*optrA* (n = 2, 14.3%)		2 (100)		
*optrA*+*cfr(D) + poxtA-EF9F6* (n = 1, 7.1%)		1 (100)		
*cfr(D) + poxtA-EF9F6* (n = 1, 7.1%)		1 (100)		
*cfr(D) + poxtA2* (n = 2, 14.3%)	1 (50.00)	1 (50.0)		
G2576T (n = 4, 28.6%)		1 (25.0)	1 (25.0)	2 (50.0)
unknown mechanism (n = 4, 28.6%)		1 (25.0)	2 (50.0)	1 (25.0)
***E. gallinarum* (n = 2)**	**0**	**2 (100)**	**0**	**0**
*poxtA-EF9F6* (n = 2, 100%)		2 (100)		
**Total (n = 34)**	**5 (14.7)**	**23 (67.6)**	**3 (8.8)**	**3 (8.8)**
*optrA* (n = 17, 50.0%)	1 (5.9)	16 (94.1)		
*optrA*+*cfr(D) + poxtA-EF9F6* (n = 1, 2.9%)		1 (100)		
*cfr(D) + poxtA-EF9F6* (n = 1, 2.9%)		1 (100)		
*cfr(D) + poxtA2* (n = 5, 14.7%)	4 (80.0)	1 (20.0)		
*poxtA-EF9F6* (n = 2, 5.9%)		2 (100)		
G2576T (n = 4, 11.8%)		1 (25.0)	1 (25.0)	2 (50.0)
unknown mechanism (n = 4, 11.8%)		1 (25.0)	2 (50.0)	1 (25.0)

## Data Availability

The data have been deposited on NCBI under the BioProject ID PRJNA892662.

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
