# Peer review of "Prevalence and Genetic Analysis of Resistance Mechanisms of Linezolid-Nonsusceptible Enterococci in a Tertiary Care Hospital Examined via Whole-Genome Sequencing"

_antibiotics, 2022, doi:10.3390/antibiotics11111624_

Round 1

Reviewer 1 Report

The manuscript investigates the prevalence of linezolid resistance mechanisms in a tertiary care hospital. The authors randomly selected 34 linezolid nonsusceptible enterococci strains for genetic analysis of linezolid resistance mechanisms.

Major Comments

Information regarding the 34 strains should be available, namely the antimicrobial profile, collection date, and clinical sample. Are these strains multidrug resistant? Regarding epidemiological data, it would also be interesting to have information regarding the previous antibiotic therapy of patients (if possible) to be aware of the antimicrobial selective pressure. Sequencing data should be publicly available (uploaded in GenBank).

Results analysis from the genetic environment of different resistance determinants were well presented, but the authors missed describing how to perform the analysis in material and methods.

The clonal relatedness analysis was disappointing. The relatedness of the different linezolid nonsusceptible enterococci was assessed but without indication of the genetic distance and clonal relation among strains, and the possibility of occurrence of transmission/contaminations was not assessed.

The quality of all figures is not satisfactory. Please provide better quality figures (at least 300 dpi, please consult the author's instructions)

Minor Comments

Abstract

In lines 13-24: You mentioned “—which were responsible for an epidemic in recent years—” . It is unusual to add references in the abstract. Therefore, I kindly ask you to provide the respective context of that sentence in the introduction and add a reference. Otherwise, reformulate the sentence in the abstract, as it is unusual to add references in the abstract.

Introduction

Line 53: “haven’t been” change to “have not been”

Material and methods

Line 286: delete “kind of”

Line 295: delete “from 2016 to 2020”

Line 289-300: please reformulate for clarity (change the order of the sentences, explain the criteria for selection and then indicate the isolates selected).

Line 304: eliminate “precise”

Line 317: Why did you mean with “novo hybrid assembly” if you performed only illumina sequencing?

Line 318-319: Authors state “Resistance gene analysis and plasmid prediction were performed using ResFinder 4.1”. However, ResFinder 4.1 does not permit plasmid prediction. Please clarify.

Line 306: Please indicate the manufacture of linezolid used to prepare agar plates according to CLSI procedures

Lines 313-317: Please add the reference for Unicycler v0.4.6. You need to upload the sequencing data on-line database (GenBank) and accession numbers should be included in the manuscript.

Results

Table 1. Please indicate in the table the MICs values of the CLSI clinical breakpoints used as interpretation criteria, as many readers do not have access to the 31st edition of the Clinical and Laboratory Standards Institute (CLSI).

Lines 94---: For a better understanding of the following results, please add a supplementary table with the information on strains ID, biological sample from which they were recovered, date, antimicrobial susceptibility testing (Phenotype), MLST, resistome (genotype), and accession number.

Line 96: Correct “with” to “in”

Line 100: eliminate “development”

Line 116- Please add information in the material and method section regarding the results from this section 2.3 “Genetic environment of different resistance determinants”, namely the blast analysis described in Results is not described in the Materials and Methods.

Line 119: “All resistance genes were located in the chromosome… “ The method used does not allow identifying with certainty the location of antimicrobial genes (genome vs plasmid). Long-reads sequencing has to be used to confirm this. Please clarify.

Line 119-120: Please add “strains” before “EFM 119 30, …”

Line 122: Please add “strains” before “EFM 30…”

Line 184. The Section “2.4. Clonal relatedness and other molecular features” was disappointing. It is not clear the relatedness of the different strains as there were no information of the genetic distance, including in the branches of the trees (Figure 4).

There authors said: The clonal relatedness between E. faecium isolates was high (all strains belonged to CC17), but no interpretation of the clonal relatedness was made. For example, strains in the same branch as EFA66 and EFA34 are clones? How are they related to the EFA25 (genetic distance). Can this strain be part of that clone? Were they recovered in the same month? Moreover, no information was provided about the presence of resistant determinants other than linezolid which could be interesting to show in the trees.

Line 194: Please explain what you mean by “discrepancies among alleles were also higher”. If the ST from E. faecium were diverse, naturally, the alleles had to be distinct.

Discussion

Line 207-208 “In Korea, relatively few reports have been published examining linezolid resistance in clinical E. faecalis isolates.” Please provide the reference(s)

Line 221-213: Please reformulate “By comparison, in our study, the proportion of linezolid-resistant E. faecium was higher, and the proportion of E. faecium strains showing intermediate susceptibility to linezolid was reduced.” Suggestion: Likewise, in our study, the frequency of linezolid-resistant E. faecium was higher than…..”

Line 215: The authors suggest that the shift from intermediary to linezolid resistance can be attributed to “….continued widespread use of linezolid”.  Please provide information regarding the trend of consumption of linezolid in Korean from the last years. Is linezolid consumption in Korea hospitals recorded?

(Data Availability Statement) Line 340: Whole genome sequence data should be deposited to the on-line database (GenBank) and the accession numbers should be included in the manuscript. You can check the Instructions for Authors. (https://www.mdpi.com/journal/microorganisms/instructions)

Reviewer 2 Report

Hu al. describe the application of whole genome sequencing to antibiotic resistance analysis of linezolid-resistant Enterococci in a hospital. In general, the manuscript is well written and easy to comprehend. Unfortunately, the claims of resistance mechanism elucidation are not scientifically sound and are not supported by the data presented. I have several comments to be addressed.

Major comments:

1. How did you determine chromosome or plasmid origin of a particular gene? There is no indication that precise plasmid sequences were reconstructed from WGS data. The location of a gene within a particular contig does not prove its plasmid origin since similar fragments can occur on both plasmid and chromosome. It is not uncommon for bacteria to have a particular resistance gene on both chromosome and plasmid at once. What was the read coverage of contigs? How can you verify that gene location on a particular contig is not an assembly artifact? Long-read sequencing or PCR can be used to verify your claims.

2. The presence of particular gene does not prove its expression and functionality. In addition, the authors have not obtained the reliable (i.e., statistically significant) data matching the gene content and resistance of the isolates. No correlation data was provided either. The involvement of a gene in providing resistance should be verified by constructing knockout clones if the mechanism of resistance was not confirmed previously. Thus the claims of having resistance mechanisms are purely speculative and are not unambiguously supported by the data.

3. Please include the table with specimen description for all isolates sequenced. These data is essential for understanding the possible genomic differences (possibly, in supplementary data)

4. According to journal policy, the sequences of genomes for the isolates studied should be deposited to public databases. Please do this if not done already and specify accession numbers in Data Availability section.

5. Please consider including the colored figures 1-4 – they will become much more informative and reader-friendly

Minor comments:

Line 12 – please clarify “after its introduction” – does it mean introduction of linezolid into clinical practice or something else?

Line 18 – please clarify that you used short-read sequencing

Line 35 – should be “antimicrobial resistance genes”. Genes cannot be resistant.

Line 54 – what do you mean by “the sore resistance mechanism”?

Line 224 – the presence of a particular gene cannot be the mechanism of resistance since the presence does not always mean that the gene is expressed and functional

Line 239 – please italicize E. faecalis

Line 295 – please indicate the CLSI version used – was it the same for all isolates during the collection period?

Line 301-302 – should be “resistance mechanisms”

Line 309 – should be “were greater” (sequence of tenses)

Line 317 – hybrid assembly involves using short and long reads in assembly process. As far as I understood, only short reads were used in this study, so the assembly is not a hybrid one

Line 319 – Resfinder does not predict plasmids. Please check.

Please change the reference list to comply with journal rules (numbers, bold font for year of publication, number of authors etc).

Round 2

Reviewer 1 Report

Thank you.

Best regards.

Author Response

Thank you for taking the time to review our manuscript. We sincerely appreciate all the suggestions and comments you gave. We will actively discuss with the editorial board to proceed with the follow-up work.

Reviewer 2 Report

Most comments have been addressed appropriately. However, I recommend adding less rigorous claims to the Abstract regarding the mechanisms of resistance (lines 20-23) since the mechanisms were not confirmed by experiments. For example, "was the likely primary resistance mechanism" or "primary predicted mechanism" etc. This will help readers to make appropriate conclusions from the data provided.

Author Response

Thank you for your comment. We have modified the Abstract to make it less rigorous. Please see Line 20-23 or below:

"Results: The presence of the optrA gene was likely the primary resistance mechanism in these three species, typically demonstrating an MIC value of 8 μg/ml. Co-existence of cfr(D) and poxtA2 gene was the second major mechanism, primarily predicting a phenotype showing intermediate susceptibility to linezolid."